# RevColV2: Exploring Disentangled Representations in Masked Image Modeling

**Qi Han**[*]
MEGVII Technology
Beijing, China
hqer@foxmail.com

**Yuxuan Cai**[*]
MEGVII Technology
Beijing, China
larryx.tsai@gmail.com

**Xiangyu Zhang**[†]
MEGVII Technology
Beijing, China
zhangxiangyu@megvii.com

## Abstract

Masked image modeling (MIM) has become a prevalent pre-training setup for vision foundation models and attains promising performance. Despite its success, existing MIM methods discard the decoder network during downstream applications, resulting in inconsistent representations between pre-training and fine-tuning and can hamper downstream task performance. In this paper, we propose a new architecture, RevColV2, which tackles this issue by keeping the entire autoencoder architecture during both pre-training and fine-tuning. The main body of RevColV2 contains bottom-up columns and top-down columns, between which information is reversibly propagated and gradually disentangled. Such design enables our architecture with the nice property: maintaining disentangled low-level and semantic information at the end of the network in MIM pre-training. Our experimental results suggest that a foundation model with decoupled features can achieve competitive performance across multiple downstream vision tasks such as image classification, semantic segmentation and object detection. For example, after intermediate fine-tuning on ImageNet-22K dataset, RevColV2-L attains 88.4% top-1 accuracy on ImageNet-1K classification and 58.6 mIoU on ADE20K semantic segmentation. With extra teacher and large scale dataset, RevColv2-L achieves 62.1 $AP_{box}$ on COCO detection and 60.4 mIoU on ADE20K semantic segmentation.

## 1 Introduction

Pre-trained vision foundation models attract more attention in vision community [1, 2, 3, 4]. A key component in pre-training is how to learn generalizable features which meet the demands of various visual applications [5]. Typical series of methods focus on self-supervised learning, such as contrastive learning [6, 7] and masked image modeling (MIM) [8, 9]. The latter obtains promising results recently in most scenarios by learning occlusion invariant features [10], and becomes a commonly used approach in vision pre-training especially for large models [11, 4, 12, 13].

---

[*]Equal contribution

[†]Corresponding author. This work is supported by The National Key Research and Development Program of China (No. 2017YFA0700800).

37th Conference on Neural Information Processing Systems (NeurIPS 2023).

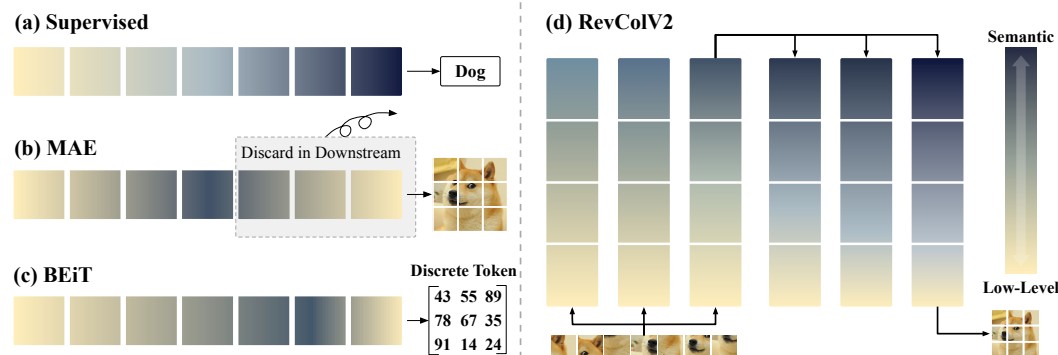

Figure 1: Illustration of the information distribution in different pre-training schemes. (a): low-level information is gradually changed to be semantic in supervised pre-training; (b) and (c): low-level information firstly changes to be semantic then recovers to be low-level or entangled in MAE and BEiT pre-training; (d): low-level and semantic information is gradually disentangled in RevColV2 architecture with MIM pre-training.

A typical MIM method, masked autoencoders (MAE) [8], employs an encoder to embed the masked images into semantic features and a decoder to reconstruct unseen patches, as shown in Figure 1 (b). Under such pre-training paradigm, features are rich in low-level information in both input and output. The semantic features, which are desired for downstream tasks, are reserved inside network. A common method to utilize such semantic feature is to manually partition the encoder and decoder based on the amount of semantic information in features and discard the decoder during downstream fine-tuning [8, 14, 15, 16]. Even so, discarding parts of the pre-trained network could incur information loss when transferring to downstream visual tasks. Existing works alleviate this defect by utilizing encoder only architectures [9, 17] and jointly modeling the un-masked patches and mask tokens. Even with extra computation cost in pre-training, these methods often show inferior generalization abilities, because the low-level information used for reconstructing images and the semantic information still appears in an entangled form during pre-training, as shown in Figure 1 (c).

In this paper, we tackle this problem in terms of architecture design. Rather than discarding the decoder, we keep the entire autoencoder architecture in both pre-training and fine-tuning. Similar architecture has proven to be successful in masked language modeling [18]. Nevertheless, vision tasks are naturally different from language tasks, due to their inconsistent output space between pre-training and fine-tuning [19, 20]. In order to obtain better transfer ability as well as keep the unified encoder-decoder architecture, it is important to separate the low-level and semantic information during the image reconstruction process in pre-training. RevCol [1] is a pioneer that combines the idea of reversible network [21, 22] and multi-column architecture [23, 24] to learn disentangled representation in label-guided supervised pre-training. A straightforward attempt is directly combining RevCol with the decoder used in MAE to perform MIM pre-training. However, in order to reconstruct raw images, both low-level and semantic information is needed for reasoning the unseen content and detail appearance, resulting in entangled information. This not only harms the downstream tasks but also destroys the disentangled learning objective of RevCol.

As shown in Figure 2, we re-design the architecture of RevCol to fit the MIM pre-training target. The new architecture contains a bottom-up reversible column encoder and a top-down reversible column decoder. The bottom-up columns and top-down columns are totally symmetric with masked images and encoder embedding as input. During MIM pre-training, the raw image reconstruction loss is connected to the end of the last column in decoder. Hence low-level information primarily sinks to the bottom level and semantic information moves upwards to other stages based on lossless propagation, as shown in Figure 1 (d). Between bottom-up columns and top-down columns, reversible connections are added to ensure the disentangle feature learning object of the whole network. Benefited from the disentangled representations, during downstream fine-tuning, features in the last top-down columns can be quickly adapted to various tasks. Thus the unified architecture does not have to discard the decoder network[3] and avoids the mixture of low-level and semantic information, fully exploring the pre-training abilities. The re-designed new architecture is named *RevColV2*.

---

[3]Our decoder behaves differently from traditional 'decoder' which refers to the network performing merely image reconstruction. The top-down reversible column decoder also includes semantic information in our design.

We build various RevColV2 models with different number of parameters and computation budgets and evaluate them on downstream ImageNet [25] image recognition, ADE20K [26] semantic segmentation and COCO [27] object detection after MIM pre-training. The experimental results show consistent improvements over RevCol(V1). Compared with other counterparts, RevColV2 achieves comparable performance to state-of-the-art models pre-trained on pure ImageNet-1K dataset, verifying the effectiveness of the new RevColV2 architecture. After intermediately fine-tuning on ImageNet-22K dataset, RevColV2-L achieves 88.4% top-1 accuracy on ImageNet-1K and 58.7 mIoU on ADE20K segmentation. Moreover, we design a new pre-training scheme which joint modeling the masked semantic features in the top-level, as well as the low-level image pixels in the bottom. Under such pre-training scheme and large scale dataset Laion400M[28], RevColv2-L achieves 62.1 AP$_{box}$ on COCO detection. In ADE20K segmentation, RevColv2-L reaches 60.4 mIoU with multi-scale test. We also conduct analytical experiments showing that RevColV2 with bottom-up reversible columns and top-down reversible columns can learn disentangled representation in MIM pre-training. This property benefits RevColV2 to share unified architecture between pre-training and fine-tuning, fully exploiting the potential of vision pre-training.

## 2 RevCol V2

In this section, we introduce the newly designed RevColV2, which is a pure isotropic transformer architecture [29]. The core idea in RevColV2, learning disentangled representations during MIM pre-training, is accomplished by the proposed symmetrical reversible encoder-decoder columns and the unified architecture in pre-training and fine-tuning.

### 2.1 Preliminary

RevCol [1] is a reversible column architecture which learns disentangled representation in supervised learning. The main body of RevCol is composed of multiple subnetworks named columns respectively. Each column contains several basic residual blocks and could be partitioned into four levels with corresponding feature maps. Between each column, reversible connections are introduced to keep lossless information propagation. In Equation 1, we summarize the forward and inverse propagation function in RevCol.

$$Forward : x_i^l = \boldsymbol{F}_i^l(x_{i-1}^l, x_{i+1}^{l-1}) + \gamma x_i^{l-1}$$
$$Inverse : x_i^{l-1} = \gamma^{-1}[x_i^l - \boldsymbol{F}_i^l(x_{i-1}^l, x_{i+1}^{l-1})], \tag{1}$$

where $x_i^l$ denotes feature maps of the $i$-th level in $l$-th column; $\boldsymbol{F}_i^l$ denotes an arbitrary non-linear operation in $i$-th level, analogous to those residual functions in standard *ResNets*; $\gamma$ is a reversible operation (*e.g.* channel-wise scaling), whose inverse is denoted by $\gamma^{-1}$. The $i$-th level takes the features of the lower level at the same column $x_{i-1}^l$ and the upper level at the previous column $x_{i+1}^{l-1}$ as input. These two features are fused together and passed through a stack of building blocks. After that, feature $x_i^{l-1}$ are added with reversible operation $\gamma$ to get the final output. Given the features within one column, we can compute the features in other columns recursively during forward and backward propagation according to Equation 1. At the back-propagation, we can reconstruct the required activations *on the fly* from the last column to the first, which means we only need to maintain activations from *one* column in memory during training.

With this reversible property, RevCol can learn disentangled representations in supervised pre-training: the most semantic information is maintained in the last level according to the class label target, and the irrelevant low-level information is kept in other levels based on lossless propagation. In RevColV2, we inherit the RevCol design paradigm but make some improvements for MIM pre-training.

### 2.2 Symmetrical Encoder and Decoder with Reversible Columns

Figure 2 gives a sketch of the RevColV2 top-level architecture design. The bottom-up column encoder follows RevCol [1] in macro design. The input image is first split into non-overlapping patches by a patch-embedding module. In MIM pre-training, the image patches are randomly masked and the un-masked patches are input into each bottom-up column. The forward and inverse function still follows Equation 1, but we change the operation inside $F$. We select hardware-friendly vanilla transformer block [29] which includes pre-LayerNorm, Self-Attention and Feed-Forward network (FFN) as the basic building blocks in each column. The blocks are evenly split into four levels for

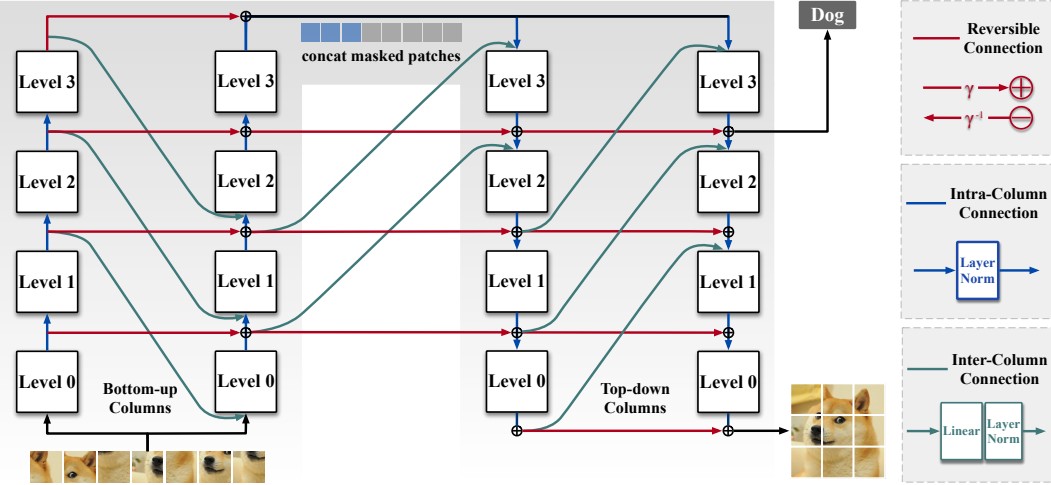

Figure 2: The pipeline of RevColV2 is a unified architecture between pre-training and fine-tuning. The bottom-up columns inherit the design of RevCol with masked image patches as inputs. In the pre-training, the top-down columns receive the outputs of bottom-up columns along with the mask tokens, and reconstruct unseen raw patches at the lowest level of the last column. In the fine-tuning, the output features of top-down columns are selectively used in downstream tasks, such as classification task using the top level features.

easy implementation of downstream tasks. Two input feature maps of each level are first normalized by a LayerNorm module and then summed together. Consider one of the input $x_{i+1}^{l-1}$ is from the previous column, we add another linear transformation on this inter-column branch to project the input space into current column's, as shown in Figure 2.

The top-down column decoder is symmetric to the bottom-up column encoder. Similar as MAE [8], output features of the last bottom-up column are first normalized and concatenated with learnable mask tokens, then input into the top-down columns. As the information propagation is in the opposite direction, the input features of each level are also opposite to the encoder. Equation 1 becomes:

$$
\begin{aligned}
Forward &: x_i^l = \boldsymbol{F}_i^l(x_{i+1}^l, x_{i-1}^{l-1}) + \gamma x_i^{l-1} \\
Inverse &: x_i^{l-1} = \gamma^{-1}[x_i^l - \boldsymbol{F}_i^l(x_{i+1}^l, x_{i-1}^{l-1})].
\end{aligned}
\tag{2}
$$

Two inputs of $\boldsymbol{F}_i^l$ becomes $x_{i+1}^l$ (feature of the upper level in current column) and $x_{i-1}^{l-1}$ (feature of the lower level in previous column). For the input (to the highest level) of each top-down column, $x_{i+1}^l$ is actually the last level's output feature of the encoder, which follows the design of image patches input to each bottom-up column. At the lowest level of the last top-down column, a pixel-level loss is add to reconstruct the unseen image patches. Obviously, the top-down column is also reversible. Apart from the input, all other calculations remain the same as the encoder, and that's why we describe the encoder and decoder as fully-symmetrical.

Compared with existing architectures [1, 29], the symmetrical reversible column architecture in MIM pre-training has the following advantages:

- The disentangled feature learning object in RevCol is no longer restricted to supervised learning. Although the image reconstruction loss at the end of decoder requires mostly low-level information, the semantic information can still be maintained in other levels and gradually refined through reversible propagation. Therefore, we can learn disentangled representations in MIM pre-training without labels.

- Low-level and semantic information is retained in both bottom-up columns and top-down columns. Thus there is no need to discard parts of the network during fine-tuning. The bottom-up encoder and top-down decoder can serve as a unified architecture that yields consistent representation during pre-training and fine-tuning.

## 2.3 The Unified Architecture between Pre-training and Fine-tuning

**MIM Pre-training.** As shown in Figure 2, in MIM pre-training, masked image patches are fed into the bottom-up columns and reconstruct unseen patches through top-down columns. At the lowest level of the last top-down column, we use mean-square error (MSE) loss to reconstruct raw images, similar to [8]. As described in Section 2.2, the low-level information is mainly gathered in the bottom levels (yellow regions in Figure 1(d)) with the constraint of MSE loss. Under the constraint of lossless propagation, the semantic information is accordingly decoupled to the top levels (blue regions in Figure 1(d)). Under such circumstances, the top-down column decoder learns not only reconstruction details, but also semantic features, indicating that downstream tasks can directly take advantage of these decoder outputs.

**Joint Pre-training.** Considering the semantic information will be gathered in the top levels during MIM training, we make a further step. We *explicitly* joint model the masked semantic features in the top-level, as well as the low-level image pixels in the bottom. Specifically, we introduce CLIP [30] as as an extra teacher to model the semantic features and use cosine similarity as loss function. We also keep the pixel reconstruction MSE loss. Thanks to the disentangled feature in the last column, we can apply two different loss separately at the top and bottom. Our method is different from raw pixel reconstruction methods like MAE [8] or mask distillation works like MaskDistill [31], which modeling homogeneous features at the output of the network.

**Downstream Fine-tuning.** Rather than discard the decoder during fine-tuning, we leverage the entire encoder-decoder architecture. We come up with two fine-tuning methods in downstream tasks, according to different characteristics. For classification tasks, which require highly semantic features only, we apply classification heads on top of the last top-down column. Since the whole network is optimized by raw pixel reconstruction pre-training target, low-level information sinks to the bottom level features and semantic information is preserved at the top. For dense prediction tasks, which require of both semantic and low-level information, we take all level's features of the last top-down column, then directly connect them to task oriented dense prediction heads.

Compared with exists autoencoder architectures [8, 14, 32, 33, 34] that have to drop the decoder in downstream applications, the unified RevColV2 architecture has consistent representations during pre-training and fine-tuning, fully taking advantage of vision pre-training. Meanwhile, encoder only architectures [9, 17, 4, 31] often entangle the low-level and semantic information in the final output. However, RevColV2 can learn disentangled representations during pre-training, naturally avoids these problems. Compared with CNN-based backbones, ViT backbones are cumbersome to transfer in dense prediction tasks, often with an adapter like ViT-Adapter [35] or SimpleFPN in ViTDet [36]. Thanks to the multi-level embedding output, RevColV2 is free from any extra feature pyramid structures (eg, FPN [37], BiFPN[38], etc) or adapters. We simply interpolate the output features in the last column to multiple resolutions, then leverage the hierarchical embedding in dense prediction tasks. The simple and effective solution for dense prediction tasks makes the ViT-based RevColV2 easy to transfer, like previous CNN-based backbones.

## 2.4 Model Variants

We provide two variants of RevColV2 models, **B**ase and **L**arge, as shown in Table 1. The depth of bottom-up columns and top-down columns are fixed to 12 and 4. The number of bottom-up columns and top-down columns is changed from 3 to 4 according to different model sizes. The model parameters include the whole bottom-up columns and top-down columns. However, in the downstream task such as image classification, not all levels are involved in computation (as shown in Figure 2), such the number of parameters is slightly lower (as shown in Table 2). We do not use intermediate supervision for each column as it needs careful tuning.

Table 1: RevColV2 architecture configurations for different model sizes. FLOPs are measured on classification task with $224^2$ resolution input. BU and TD are short for bottom-up and top-down.

|  | #BU columns. | BU depth | #TD columns. | TD depth | dim | head dim | Params. | FLOPs |
|---|---|---|---|---|---|---|---|---|
| Base | 3 | 12 | 3 | 4 | 416 | 32 | 101M | 19G |
| Large | 4 | 12 | 4 | 4 | 672 | 56 | 342M | 67G |

Table 2: ImageNet-1K classification results for various sizes models. The left table shows the end-to-end fine-tune results after MIM pre-training, and the right table shows the results after intermediate fine-tuning on ImageNet-22K.

| Model | Size | Target | Params | FLOPs | FT |
|---|---|---|---|---|---|
| ***ImageNet-1K pre-train:*** | | | | | |
| BEIT-B [9] | $224^2$ | DALL-E | 87M | 18G | 83.2 |
| MAE-B [8] | $224^2$ | Pixel | 87M | 18G | 83.6 |
| CAE-B [14] | $224^2$ | DALL-E | 87M | 18G | 83.9 |
| SdAE-B [40] | $224^2$ | EMA | 87M | 18G | 84.1 |
| MaskFeat-B [32] | $224^2$ | HOG | 87M | 18G | 84.0 |
| ConvNeXt-B [41] | $224^2$ | Pixel | 89M | 15G | 83.8 |
| SimMIM-B [17] | $224^2$ | Pixel | 88M | 16G | 84.0 |
| HiViT-B [42] | $224^2$ | Pixel | 66M | 16G | 84.2 |
| DeiT III-B [43] | $224^2$ | Label | 87M | 18G | 83.8 |
| HorNet$_{GF}$-B [44] | $224^2$ | Label | 88M | 16G | 84.3 |
| SwinV2-B [11] | $224^2$ | Label | 88M | 20G | 84.2 |
| ConvNeXt V2-B [12] | $224^2$ | Pixel | 89M | 15G | 84.9 |
| RevCol-B [1] | $224^2$ | Label | 138M | 17G | 84.1 |
| RevColV2-B | $224^2$ | Pixel | 88M | 19G | **84.7** |
| DeiT III-L [43] | $224^2$ | Label | 304M | 62G | 84.9 |
| BEiT-L [9] | $224^2$ | Pixel | 307M | 62G | 85.2 |
| MAE-L [8] | $224^2$ | Pixel | 307M | 62G | 85.9 |
| CAE-L [14] | $224^2$ | DALL-E | 307M | 62G | 86.2 |
| MaskFeat-L [32] | $224^2$ | HOG | 307M | 62G | 85.7 |
| SimMIM-L [17] | $224^2$ | Pixel | 197M | 35G | 85.4 |
| ConvNeXt V2-L [12] | $224^2$ | Pixel | 198M | 34G | 85.8 |
| RevColV2-L | $224^2$ | Pixel | 327M | 67G | **86.3** |

| Model | Size | Target | Params | FLOPs | FT |
|---|---|---|---|---|---|
| ***ImageNet-1K pre-train + 22K intermidate fine-tune:*** | | | | | |
| ViT-B [41] | $384^2$ | Label | 86M | 55G | 84.0 |
| DeiT III-B [43] | $224^2$ | Label | 87M | 18G | 85.7 |
| ConvNeXt-B [41] | $224^2$ | Pixel | 89M | 15G | 85.8 |
| ConvNeXt-B [41] | $384^2$ | Pixel | 89M | 45G | 86.8 |
| RevCol-B [1] | $224^2$ | Label | 138M | 17G | 85.6 |
| RevCol-B [1]↑ | $384^2$ | Label | 138M | 49G | 86.7 |
| RevColV2-B | $224^2$ | Pixel | 88M | 19G | **86.2** |
| RevColV2-B↑ | $384^2$ | Pixel | 88M | 64G | **87.3** |
| RevColV2-B↑ | $512^2$ | Pixel | 88M | 130G | **87.5** |
| ViT-L [29] | $384^2$ | Label | 307M | 191G | 85.2 |
| DeiT III-L [43] | $224^2$ | Label | 304M | 62G | 87.0 |
| MOAT-3 [45] | $224^2$ | Label | 190M | 45G | 86.8 |
| SwinV2-L [11] | $256^2$ | Label | 197M | 48G | 86.9 |
| SwinV2-L↑ [11] | $384^2$ | Label | 197M | 115G | 87.6 |
| ConvNeXt V2-L [12] | $224^2$ | Pixel | 198M | 34G | 87.3 |
| ConvNeXt V2-L↑ [12] | $384^2$ | Pixel | 198M | 103G | 88.2 |
| RevCol-L [1] | $224^2$ | Label | 273M | 39G | 86.6 |
| RevCol-L↑ [1] | $384^2$ | Label | 273M | 116G | 87.6 |
| RevColV2-L | $224^2$ | Pixel | 327M | 67G | **87.4** |
| RevColV2-L↑ | $384^2$ | Pixel | 327M | 215G | **88.3** |
| RevColV2-L↑ | $512^2$ | Pixel | 327M | 417G | **88.4** |

# 3 Experiments

## 3.1 MIM Pre-training

### 3.1.1 Pre-training Details

We pre-train RevColV2 on ImageNet-1K [25] dataset. Hyper-parameters generally follow [8]. The mask ratio is set as 75% with random sampling strategy and the reconstruction target is the normalized raw pixel from the original image. We pre-train 1600 epochs for RevColV2 models. The pre-training image size is $224^2$ and the pre-training optimization parameters are: batch size 4096, base learning rate 1.5e-4 for 256 batch-size and linear scaled up, AdamW with weight decay 0.05. We do not use stochastic depth strategy in pre-training. More details can be found in supplementary material.

### 3.1.2 ImageNet Classification

**Setup.** For image classification, we evaluate top-1 accuracy on ImageNet-1K [25]. We initialize weights using MIM pre-trained models, and fine-tune on ImageNet-1K with class label, similar to [8, 12, 9]. To fully exploit the potential of RevColV2, we intermediately fine-tune the models on ImageNet-22K [39] following [34, 12] after MIM pre-training. The intermediate fine-tuning hyper-parameters are almost the same as [12] and shown in supplementary material.

**Results.** Table 2 shows the results on ImageNet-1K classification. The MIM pre-train epochs for RevColV2 is 1600, and other methods are reported with their longest schedules. RevColV2-B achieves **84.7** top-1 accuracy with pure ImageNet-1K data, outperforms ViT [29] architecture pre-trained with MIM [8, 9, 14, 40, 32] by a large margin using only raw pixels as reconstruction target. As a pure transformer isotropic architecture, RevColV2-B also achieves comparable performance with state-of-the-art hierarchical architectures, *i.e.* RevColV2-B reaches higher performance than SwinV2-B[11], HorNet$_{GF}$-B [44] and other counterparts in Table 2. For large size models, RevColV2-L with **86.3%** top-1 accuracy outperforms ConNeXt V2 [12], MAE [8], and CAE [14] counterparts.

We initialize RevColV2 models with the MIM pre-trained model weights and then use larger dataset ImageNet-22K and supervised training methods to test the scaling up ability. Under this training

Table 3: Semantic segmentation result on ADE20K dataset with UperNet and Mask2Former framework. *M2F* denotes using Mask2Former framework. We report mIoU with single/multi-scale test.

| Backbone | mIoU (ss.) | mIoU (ms.) | Params | Backbone | mIoU (ss.) | mIoU (ms.) | Params |
|---|---|---|---|---|---|---|---|
| *ImageNet-1K:* | | | | *ImageNet-1K + ImageNet-22K:* | | | |
| MAE-B [8] | 48.1 | N/A | 156M | Swin-B [49] | 50.3 | 51.7 | 121M |
| CAE-B [8] | 50.2 | N/A | 156M | ConvNeXt-B [41] | 52.6 | 53.1 | 122M |
| PeCo-B [48] | 48.5 | N/A | 156M | RevCol-B [1] | 52.7 | 53.3 | 122M |
| Swin-B [49] | 48.1 | 49.7 | 121M | RevColV2-B | **53.1** | **53.9** | 121M |
| ConvNeXt-B [41] | 49.1 | 49.9 | 122M | RevColV2-B+*M2F* | **54.9** | **55.8** | 325M |
| ConvNeXt V2-B [12] | N/A | 52.1 | 122M | Swin-L [49] | 52.1 | 53.5 | 234M |
| InternImage-B [50] | 50.8 | 51.3 | 128M | ConvNeXt-L [41] | 53.2 | 53.7 | 235M |
| RevCol-B [1] | 49.0 | 50.1 | 122M | RepLKNet-L [51] | 52.4 | 52.7 | 207M |
| RevColV2-B | **51.3** | **52.3** | 121M | Focal-L [52] | 54.0 | 55.4 | 240M |
| BEiT [9] | 53.3 | N/A | 421M | CSwin-L [53] | 54.0 | 55.7 | 208M |
| MAE-L [8] | 53.6 | N/A | 421M | RevCol-L [1] | 53.4 | 53.7 | 306M |
| ConvNeXt V2-L [12] | 53.2 | 53.7 | 235M | RevColV2-L | **54.8** | **55.7** | 399M |
| RevColV2-L | **54.0** | **54.4** | 399M | RevColV2-L+*M2F* | **58.2** | **58.6** | 570M |

Table 4: Object detection and instance segmentation results on COCO dataset. † means using ViT-Adapter [35] networks. †† means that model pre-trained on ImageNet-22K dataset and ⚗ denotes using additional distillation teacher in pre-training.

| Backbone | $AP_{box}$ | $AP_{mask}$ | Params | Backbone | $AP_{box}$ | $AP_{mask}$ | Params |
|---|---|---|---|---|---|---|---|
| *with Mask R-CNN:* | | | | *with Cascade Mask R-CNN:* | | | |
| MAE-B [8] | 49.8 | 44.3 | 110M | Swin-B [49] | 51.9 | 45.0 | 145M |
| MAE-B† [8] | 51.3 | 45.4 | 122M | ConvNeXt-B [41] | 52.7 | 45.7 | 146M |
| RevColV2-B | **52.4** | **46.2** | 119M | RevCol [1] | 53.0 | 45.9 | 196M |
| MAE-L [8] | 53.1 | 47.1 | 323M | ViTDet-B [36] | 54.0 | 46.7 | 141M |
| MAE-L† [8] | 53.3 | 47.4 | 330M | EVA-02-B †† ⚗ [54] | 55.5 | 47.1 | 141M |
| RevColV2-L | **54.0** | **47.8** | 363M | RevColV2-B | **55.2** | **47.9** | 156M |

schedule, RevColV2-B reaches **87.5%** top-1 accuracy on ImageNet-1K with $512^2$ input resolution. For large size models, the performance of RevColV2 achieves **88.4 %** top-1 accuracy, surpassing hierarchical counterparts of SwinV2 [11] and ConvNeXt V2 [12]. Compared with over RevCol(V1) [1], all sizes RevColV2s show consistent and significant improvements on classification tasks, further illustrating the effectiveness of our design.

### 3.1.3 Semantic Segmentation

**Setup.** For semantic segmentation tasks, we evaluate RevColV2 backbones on ADE20K benchmarks [26] with *UperNet* [46] and *Mask2Former* [47] framework. We interpolate the position embedding to a fixed input size for both bottom-up and top-down columns. Following previous works [11, 12], we initialize weights using ImageNet-1k classification fine-tuned models. Training hyper-parameters are available in supplementary material.

**Results.** Table 3 shows the head-to-head comparison results on ADE20K semantic segmentation with various backbones and UperNet segmentation head. RevColV2 models gain competitive performance over single-scale and multi-scale mIoU across different pure transformer and CNN architectures. RevCol-B/L pre-trained on ImageNet-1K achieve **52.3** and **54.4** mIoU on ADE20K with $512^2$ input resolution respectively, outperforming other counterparts such as ConvNeXt V2 [12]. After intermediate fine-tuning on ImageNet-22K, RevColV2-B/L reach **53.9** and **55.7** mIoU on ADE20K benchmark with $640^2$ resolution, exceeding other competitors such as Focal transformer [52] and CSwin [53]. With stronger segmentation framework Mask2Former [47], our RevCol-B/L backbones achieve **55.8** and **58.6** mIoU, further demonstrating the effectiveness of RevColV2 backbones.

### 3.1.4 Object Detection

**Setup.** For object detection and instance segmentation task, we evaluate RevColV2 backbones on COCO [27] dataset with *Mask R-CNN* [55] and *Cascade Mask R-CNN* [56] framework. We follow

Table 5: Results of the data scaling. RevColV2-L with larger pre-training data and additional teacher achieves comparable performance than 2.1G parameter RevColV1-H and EVA-02-L.

| Model | Params | Dataset | | COCO (w/o Obj365) | | | ADE20K | | |
| --- | --- | --- | --- | --- | --- | --- | --- | --- | --- |
| | | Data | teacher | Detector | $AP_{box}$ | $AP_{mask}$ | Segmenter | mIoU | +ms |
| MaskDistill-L | 0.3 G | ImageNet1K | CLIP | - | - | - | UperNet | 56.5 | - |
| BEiTv2-L | 0.3 G | ImageNet22K | CLIP | - | - | - | UperNet | 57.5 | - |
| EVA-02-L | 0.3 G | merged 33M | EVA01-CLIP | Cascade | 62.3 | 53.8 | UperNet | 60.1 | - |
| RevCol-H | 2.1 G | private 168M | semi-labeled | HTC++ | 61.1 | 53.0 | Mask2Former | 60.4 | 61.0 |
| RevColV2-L | 0.3 G | Laion400M+IN-1K | OpenCLIP | Cascade | 62.1 | 53.2 | Mask2Former | 59.5 | 60.4 |

the setting in [36] that using window attention in RevColV2 backbones. We train models with $1024^2$ resolution crops using large scale jittering augmentations [57].

**Results.** Table 4 left shows the experiment results for RevColV2 backbones compared with MAE [8] baseline using Mask R-CNN detector. For MAE baseline, we reproduce the results using the same training configuration optionally with ViT-Adapter [35]. RevColV2 series achieve **52.4** and **54.0** box AP for base and large models, outperforming MAE series. In Table 4 right, we compare RevColV2 backbones with other architectures using Cascade Mask R-CNN detector. RevColV2-B achieve **55.2** box AP and **47.9** mask AP, outperforming RevCol(V1)-B [1], ViTDet-B [36] and other competitors.

Note that in all dense prediction tasks, we do not use any extra feature pyramid structures (eg, FPN, BiFPN, etc.) or adapters like ViT-Adapters in RevColV2 backbones implementation.

## 3.2 Joint Pre-training and Data Scaling

### 3.2.1 Pre-training Details

In joint pre-training, we use OpenCLIP-L as the teacher to represent the semantic features similar to MaskDistill and EVA. Except for the additional teacher, we use a larger dataset Laion400M[28], which contains about 400M unlabeled images in pre-training. Note that we do not use datasets such as COCO, ADE20K, Object365, etc. in pre-training to avoid artificial fitting to specific distribution (this is different from EVA-02 which uses a merged dataset that has overlapped data in the downstream task). We use 800 ImageNet-1k epochs on Laion400M dataset and then 300 epochs on ImageNet-1k dataset during pre-training.

### 3.2.2 Results

Then we evaluate our model on downstream tasks such as ImageNet1K classification, COCO detection with cascade Mask-RCNN, and ADE20K semantic segmentation with Mask2Former. The newly trained RevColv2-L achieves 87.7% Top-1 accuracy in ImageNet-1k classification with $224 \times 224$ input resolution. The larger dataset and the extra teacher lead to better performance compared with purely IN-1k MIM pre-training (86.3%) and IN-1k MIM + IN-22k intermediate fine-tuning (87.4%). As shown in Table 5, the performance gain is more prominent on dense prediction tasks. RevColv2-L achieves **62.1** box AP and **52.3** mask AP in COCO instance detection and segmentation. In ADE20K segmentation, RevColv2-L reaches **60.4 mIoU** with multi-scale test.

## 3.3 Analysis

Experiments in this section are based on basic MIM pre-training.

**RevColV2 can learn disentangle representation.** Linear probing is a useful tool to evaluate the sparsity of features. We assume the semantic information tends to be more sparse, while the low-level information is more abundant. So, we evaluate the linear probing accuracy of each level to visualize the distribution of semantic and low-level information. We construct two experiments: 1) RevCol(V1) [1]+MAE [8] baseline in which the RevCol(V1) encoder has 3 columns and MAE decoder has 8 ViT blocks; 2) RevColV2 with 3 bottom-up columns encoder and 3 top-down columns decoder in which each column contains 12 blocks. The number of model parameters is maintained at the same level. Figure 3 shows the evaluation results on ImageNet-1K. The output level of the baseline encoder only reaches 47.2% accuracy, which is far less than the previous level 60.1%. The

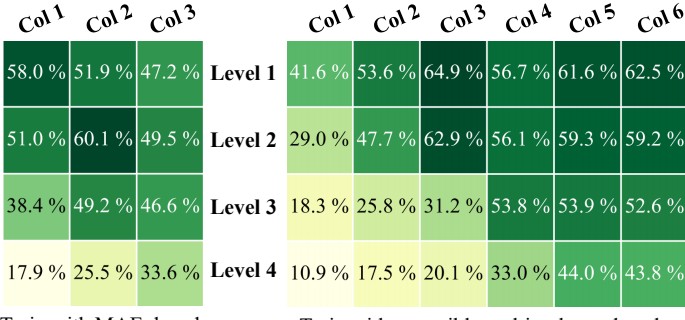

Figure 3: Linear probing accuracy of each levels on ImageNet-1K after pre-training 300 epochs. The left is RevCol+MAE baseline, and the right is RevColV2 with reversible multi-column decoder.

baseline method will entangle the low-level and semantic information during column propagation, because the MAE decoder needs both low-level and semantic features to reconstruct unseen patches. Conversely, RevColV2 with reversible top-down columns can learn the disentangled representation as described in Section 2. As shown in Figure 3, the last level of encoder and decoder reaches high linear probing accuracy (64.9% and 62.5%), and the accuracy decreases at the lower levels. Thus low-level and semantic information lies in a disentangled manner between the bottom and top levels in the last column.

We also show the linear probing accuracy of various sizes RevColV2 models in supplementary material. The separated semantic information significantly boosts the linear probing accuracy compared to other encoder only methods such as SimMIM [17].

**RevColV2 matters in unified representation fine-tune.** We experimentally verify the effectiveness of RevColV2 with unified architecture between pre-training and fine-tuning. We first compare the bottom-up encoder only variant and the entire autoencoder variant of RevColV2 on ImageNet-1K. The latter has about **0.9%** (83.8% *v.s.* 84.7%) top-1 accuracy gain, showing the top-down columns are critical in downstream fine-tuning. To eliminate the influence of model capacity (#params. and FLOPs), we also build one experiment with the same number of parameters and FLOPs for comparison. The experimental results show the accuracy of this encoder only variant (83.9%) is still lower than the entire auto-encoder variant(84.7%).

Similarly, we conduct experiments for ViT [29] + MAE [8] that without dropping decoder in the fine-tuning stage. The accuracy on ImageNet-1K drops about **0.4%** (from 83.6% to 83.2%). Give more model capacity does not increase the performance, indicating that the vanilla ViT decoder is useless in downstream fine-tuning.

We also make an ablation that only utilize the encoder's pre-trained weights while initializing the decoder weights randomly with RevColV2-B. This variant achieves 84.4% (-0.3%) top-1 accuracy on ImageNet-1K dataset and 50.7 (-0.6) mIoU on ADE20K dataset, with only ImageNet-1K MIM pre-trained encoder weights. These experimental results draw the same conclusion that the pre-trained decoder is necessary for RevColV2.

## 3.4 Ablation Study

Experiments in this section are based on basic MIM pre-training.

**Depth of top-down decoder columns** is analyzed similar to the decoder depth analysis in MAE [8], as shown in Table 6 (a). In MAE [8], the fine-tuning accuracy does not vary much over the decoder depth. However, the performance of RevColv2 drops with deeper decoder. This is because the top-down column decoder also takes up model parameters as we do not discard decoder during fine-tuning. We keep the total parameter the same, and deeper decoder will lead to shallower (narrower) encoder, which could give rise to degraded performance. This experiment also explains why we use shallower decoder (4 blocks decoder compared to 12 blocks encoder) in our RevColV2 variants.

**Different type of decoder**. The decoder in RevColV2 is a symmetric architecture with encoder, that includes multiple reversible top-down columns. To verify the effectiveness of these top-down columns, we compare this design with other variants: MAE decoder with a stack of ViT blocks; single

Table 6: Ablation results on ImageNet-1K. In different configurations of architectures, we keep the same overall FLOPs for fair comparison unless special descriptions.

(a) **Decoder column depth**.

| Depth | $ft_{Base}$ | $ft_{Large}$ |
|---|---|---|
| 4 | **84.1** | **85.2** |
| 8 | 83.6 | 84.9 |
| 12 | 82.9 | - |

(b) **Different scheduels**.

| Epochs | $ft_{Base}$ | $ft_{Large}$ |
|---|---|---|
| 300 | 84.1 | 85.2 |
| 800 | 84.5 | 85.9 |
| 1600 | **84.7** | **86.3** |

(c) **Type of decoder**.

| Decoder type | ft |
|---|---|
| MAE | 83.5 |
| UNet | 83.4 |
| Columns (w/o rev.) | 83.7 |
| Columns (w/ rev.) | **84.1** |

column decoder with shortcuts from encoder similar to UNet [58]; top-down multi-columns without shortcuts and reversible connections. Experiment results in Table 6 (c) draw the same conclusion with Section 2 that reversible column decoder achieves the best performance. This is because top-down reversible multi-columns can help feature disentangling while others not, as described in Section 2 and the analysis in Section 3.3.

**Different pre-training schedules**. We also conduct experiments with shorter schedules compared with the default 1600 epochs training, *i.e.* 300/800 epochs. Table 6 (b) shows the fine-tuning top-1 accuracy on ImageNet-1K. The results indicate larger models may need longer iterations (+0.6% from 300 to 1600 epochs for base size model, +1.1% from 300 to 1600 epochs for large size model) in MIM pre-training to fully activate the potential of models.

## 4 Related Works

**Disentangled feature learning.** Disentangled representation refers to separating the factors of variantion [5]. Traditional methods [59, 60, 61] mainly focus on generative models to learn disentangled representation. Hinton proposes GLOM [23] to build a general network that learning the part-whole hierarchies. Cai *et al.* [1] integrate the idea of GLOM with reversible networks [22, 21] to learn disentangle representation in supervised learning paradigm, namely RevCol. Although RevCol can learn disentangled representation, it needs large-scale labeled data to maintain this property that can not directly benefit from MIM. In this paper, We seek the advances of re-design RevCol with MIM to further enhance the representation abilities while maintaining the decoupled feature learning ability.

**Masked image modeling.** Self-supervised learning has a long history in computer vision research community [62, 63, 64, 65, 66, 7]. The recent method masked image modeling (MIM) erases the masked image patches and then predicts the unseen contents, a representative of which is masked autoencoders [8]. Following MIM pipeline, researchers make efforts on designing different reconstruction targets, such as DALL-E [9, 14], HOG [32], VQGAN [48], frequency [33, 67], to learn occlusion invariant features [10].

Some methods try to use consistent architecture during pre-training and fine-tuning, such as SimMIM [17] which utilizes an encoder only network. As a result, low-level and semantic information is entangled at the end of the network, resulting in degraded performance, especially for the linear probing accuracy. The above methods usually directly adopt vanilla ViT as backbones, ignoring the mutual promotion between architectures and MIM pipelines. Recent works [12, 68] aim to co-design the architecture with MIM, and focus on the CNN architectures [41]. In this paper, we explore the new research direction: learning disentangled representation during MIM. We design a unified architecture between pre-training and fine-tuning based on RevCol [1] to learn disentangled features and naturally avoid the above information mixture problem.

## 5 Conclusion

In this paper, we design a new architecture named RevColV2 which learns disentangled representations during MIM pre-training and keep a unified autoencoder architecture when transferring to downstream tasks. RevColV2 extends the reversible columns network which is previously limited in supervised learning to MIM training, and bridges the gap between pre-training and fine-tuning. These unified architectures improve the performance across various downstream tasks, including image classification, object detection and semantic segmentation, without using additional task specific adapters. By these impressive results, we hope to stimulate more research in learning generalizable features and help foundation model pre-training.

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

# A    Analysis of Speed and Memory cost

**Inference.**    The current model variants of RevColV2 introduce more latency compared with other works of the similar number of parameters and FLOPs, such as ViT. We test the inference latency of variant models in Table 7. As described RevColV1 [1], fragmented memory access takes a large part of latency. In RevColV2, we make some improvements: 1) remove the up-sample and down-sample operation in RevColV1; 2) reduce the number of total blocks; 3) hard-ware friendly architecture without hierarchy. As shown in Table 7, RevColV2 has lower latency than the V1 version during inference, but is still 1.21x higher than ViT. This is because of the large number of building blocks in RevColV2-L (about twice of ViT-L). Though we reduce the total number of blocks, the multi-column RevColv2 still requires at least 12 blocks in each column in the encoder. Shallower column leads to coarse representation which could harm the performance. On the other hand, if we make the ViT model deeper and maintain the same FLOPs, ViT-L-deeper (48 blocks) and RevColV2-L (48 blocks) have similar latency.

Table 7: Results of the inference latency, all models are tested with batch-size 32 on single A100 GPU.

| Model | FLOPs | Latency | Model | FLOPs | Latency |
|---|---|---|---|---|---|
| RevCol-L | 39G | 61 ms | RevColV2-L | 67G | 51 ms |
| ViT-L | 62G | 42 ms | ViT-L (48 blocks) | 64G | 52 ms |

Table 8: The throughput under different inference batch-size of RevColV2.

|  | bs=16 | bs=32 | bs=64 | bs=128 | bs=256 | bs=512 |
|---|---|---|---|---|---|---|
| MAE-L [8] | 730 | 754 | 786 | 811 | 820 | 823 |
| RevColV2-L [8] | 432 | 629 | 661 | 697 | 721 | 741 |
| Speedup | 0.591 | 0.834 | 0.841 | 0.859 | 0.879 | 0.900 |

We also analyze the impact of batch size. We show throughput (#image/s) under the different batch size of RevColV2-L and ViT-L on a single A100 GPU. The results in Table 8 show that with the increase in batch size, the inference speed gap between RevCol-L and ViT-L is closing because the fragment memory access time can be distributed to each sample. Although the speed of RevColV2 is lower than vanilla ViT, we think it can be solved by advanced techniques.

**Training**    We make some further analysis on the per-training speed and its memory cost, and compare them with the popular used ViT-MAE baseline. We take RevColV2-B and ViT-B for comparisons. We test training speed and memory cost on a single A100 (80GB) x 8 machine, with the same data-loader (implemented for our cluster). We use our own implementation for RevColV2 and the official implementation for MAE. Table 9 shows the training cost with batch size 4096 for one epoch. To speed up training and save memory, we equip RevColV2 with Flash Attention. We only use data parallel in this testing.

Table 9 shows that the vanilla implementation of RevColV2 pre-training has a little slower (249s vs. 220s) than ViT. Equipped with FlashAttn, RevColV2 achieves comparable pre-training cost (211s vs. 220s and 42G vs. 43G). We further analyze the impact of reversible propagation. We test the pre-training cost of the Reversible version of RevColV2 (re-compute the intermediate features during backward according to the last column outputs, rather than the vanilla autograd function in PyTorch. It is the key component of reversible column networks [1]). Results show that RevColV2-B can use extremely few GPU memory (only 18G) during pre-training with a total batch size 4096. This allows RevColV2 can be pre-trained with limited resources, such as RTX3090 GPU.

In addition to the above comparison, the fragmented access of memory can be optimized by some techniques which can be further investigated in further work. Here, we give two ways that may be further studied:

- Kernel fusion. This can reduce the frequent access of the memory caused by a large number of blocks.

Table 9: The training time and memory cost comparison on a single A100 (80GB) x 8 machine.

| Model | Time Cost | Memory (each GPU) |
|---|---|---|
| MAE-B [8] | 220 s/epoch | 43G |
| RevColV2-B [8] | 249 s/epoch | 49G |
| RevColV2-B + FlashAttn [8] | 211 s/epoch | 42G |
| RevColV2-B + FlashAttn + Reversible [8] | 240 s/epoch | 18G |

- Model parallel. Before the calculation of previous columns is finished, parts of the current column can be calculated in parallel. This is the nature of the multi-column network and can be further studied to speed up the inference and training.

## B   More Training Details

This section gives more training details on MIM pre-training and fine-tuning on downstream tasks, such as ImageNet classification, COCO detection, and ADE20K segmentation. For ImageNet experiments, the base learning rate is based on batch size 256.

### B.1   Training Details on MIM pre-training.

We use the same setting for different sizes RevCol models on MIM pre-training. The detail hyper-parameters are shown in Table 10. Following exists works [8, 12], we do not use stochastic depth [69] and other regularization strategies in MIM pre-training.

| config | value |
|---|---|
| optimizer | AdamW |
| base learning rate | 1.5e-4 |
| weight decay | 0.05 |
| optimizer momentum | $\beta_1, \beta_2 = 0.9, 0.95$ |
| batch size | 4096 |
| learning rate schedule | cosine decay |
| warmup epochs | 40 |
| training epochs | 1600 |
| augmentation | RandomResizedCrop |

Table 10: MIM Pre-training settings.

### B.2   Details on Image22K intermediate fine-tuning.

We further intermediately fine-tune RevColV2 models on ImageNet-22K dataset. The fine-tuning details is shown in Table 11. The hyper-parameters generally follow [1, 12].

### B.3   End-to-end fine-tuning details on ImageNet-1K.

We end-to-end fine-tune RevCol variants on ImageNet-1K after MIM pre-training and intermediately fine-tuning on ImageNet-22K. Table 12 shows the detail training settings after MIM pre-training.

We also show training settings on ImageNet-1K after ImageNet-22K fine-tuning. Table 13 gives the detailed hyper-parameters.

### B.4   Details on ADE20K semantic segmentation

For semantic segmentation, we evaluate different backbones on ADE20K dataset. We fine-tune the pre-trained networks on ADE20K with 160,000 iterations. For UperNet framework [46], the learning rate is 4e-5 with batch size 16, using AdamW optimizer. The layer-wise learning rate decay rate is set as 0.65 for both base and large size models. The drop path rate is 0.1. For Mask2Former

| config | value |
|---|---|
| optimizer | AdamW |
| base learning rate | 2.5e-4 |
| weight decay | 0.05 |
| optimizer momentum | $\beta_1, \beta_2{=}0.9, 0.999$ |
| layer-wise lr decay | 0.8 |
| batch size | 4096 |
| learning rate schedule | cosine decay |
| warmup epochs | 5 |
| training epochs | 90 |
| augmentation | RandAug (9, 0.5) |
| label smoothing | 0.1 |
| mixup | 0.8 |
| cutmix | 1.0 |
| drop path | 0.1 (B), 0.2 (L) |
| head init | 0.001 |
| ema | None |

Table 11: End-to-end IN-22K intermediate fine-tuning settings.

| config | value |
|---|---|
| optimizer | AdamW |
| base learning rate | 5e-4 |
| weight decay | 0.05 |
| optimizer momentum | $\beta_1, \beta_2{=}0.9, 0.999$ |
| layer-wise lr decay | 0.75 |
| batch size | 1024 |
| learning rate schedule | cosine decay |
| warmup epochs | 5 |
| training epochs | 100 (B), 50 (L) |
| augmentation | RandAug (9, 0.5) |
| label smoothing | 0.1 |
| mixup | 0.8 |
| cutmix | 1.0 |
| drop path | 0.1 |
| head init | 0.001 |
| ema | 0.9999 |

Table 12: End-to-end ImageNet-1K fine-tuning settings

framework [47], the learning rate is 2e-5 with batch size 16. The drop path rate is set as 0.3 and the layer-wise learning decay rate is 0.9.

## B.5 Details on COCO object detection and instance segmentation

For object detection and instance segmentation, we evaluate RevColV2 backbones with Mask R-CNN [55] and Cascade Mask R-CNN [56] detectors. We use ImageNet-1K MIM pre-trained weights as initialization and fine-tune the models with 50 epochs and a batch size of 32, learning rate 1e-4 for Mask R-CNN framework. The large scale jittering data augmentation strategy is used with scale range [0.1, 2.0]. The drop path rates for RevCOlV2 are set as 0.2 (base) and 0.3 (large) and the layer-wise learning rate decay rates are set as 0.9. For Cascade Mask R-CNN framework, we train models with 100 epochs following [36] with large scale jittering augmentation strategy. The learning rate is 1e-4 with batch size 64. We do not use soft-NMS in our experiments.

| config | value |
|---|---|
| optimizer | AdamW |
| base learning rate | 2.5e-5 |
| weight decay | 0.01 |
| optimizer momentum | $\beta_1, \beta_2$=0.9, 0.999 |
| layer-wise lr decay | 0.9 |
| batch size | 512 |
| learning rate schedule | cosine decay |
| warmup epochs | None |
| training epochs | 30 |
| augmentation | RandAug (9, 0.5) |
| label smoothing | 0.1 |
| mixup | None |
| cutmix | None |
| drop path | 0.1(B), 0.2 (L) |
| head init | 0.001 |
| ema | 0.9999 |

Table 13: End-to-end ImageNet-1K fine-tuning settings (after IN-22K intermediate fine-tuning).

## C   More Results

### C.1   Compared with supervised baseline

To verify the effectiveness of RevColV2 architecture with MIM pre-train, we compare the performance on ImageNet-1K fine-tune using MIM pre-trained model weights and random initialization. We use the same setting with [8] in this supervised baseline, except additional 0.999 EMA strategy. The base/large models achieve 83.1% and 82.6% top-1 accuracy on ImageNet-1K. The MIM pre-trained RevColV2 models outperform supervised baseline by a large margin (**+1.6%** and **+3.7%**).

### C.2   Linear probing results

We report the linear probing results on ImageNet-1K after pre-training for RevColV2 models and other counterparts on Table 14. Following [8], we fix the pre-trained backbone models and train a classification head for 90 epochs with LARS optimizer. We append this classification head on the last level of bottom-up columns in RevColV2. The linear probing performance of RevColV2 models surpasses other encoder only models such as SimMIM [17] and autoencoder models such as MAE [8].

Table 14: Linear probing results on ImageNet-1K dataset.

| Model | Size | Target | Params | FLOPs | LIN |
|---|---|---|---|---|---|
| ***ImageNet-1K pre-train:*** | | | | | |
| BEIT-B [9] | $224^2$ | DALL-E | 87M | 18G | 56.7 |
| SimMIM-B [17] | $224^2$ | Pixel | 88M | 16G | 56.7 |
| RevColV2-B | $224^2$ | Pixel | 88M | 19G | **64.3** |
| MAE-L [8] | $224^2$ | Pixel | 307M | 62G | 75.8 |
| RevColV2-L | $224^2$ | Pixel | 327M | 67G | **77.2** |

