# OpenReview forum: "RevColV2: Exploring Disentangled Representations in Masked Image Modeling"
_NeurIPS.cc/2023/Conference — NeurIPS 2023 poster_

### Official Review · Reviewer_W68W · 2023-07-03

**Soundness:** 4 excellent
**Presentation:** 4 excellent
**Contribution:** 3 good
**Rating:** 6
**Confidence:** 5

**Summary:**

In this paper, the authors propose a new backbone RevColv2, which is suitable to the MIM pretraining and could learn disentangled representations during the pretraining. The strong experiment results show its effectiveness.

**Strengths:**

Please refer to Questions

**Weaknesses:**

Please refer to Questions

**Questions:**

### strength
1. The paper is well-written and easy to follow
2. The proposed method is a novel combination of Mask Image Modeling and RevCol
3. The experiment part shows strong results.

### weakness
1. Some details are unclear, line 103 says 'the un-masked patches are input into each bottom-up column'. While line 134 and Fig.2 say 'masked image patches are fed into the bottom-up columns and reconstruct unseen patches through top-down column'.
Which one is the correct way?

2. Compared with vanilla ViT, the proposed model is quite complex, so I'm wondering about the speed of both the pretrain and inference.

3. Fig.3 compare the MAE decoder with RevColv2 decoder and claims that the proposed decoder can disentangle the low-level and high-level feature better. But for the linear accuracy difference between Level1 and Level4, the difference of MAE is 14.6, while the proposed method is 18.7, the difference seems not significant.

4. PeCo is a strong ImageNet-1k classification baseline while only segmentation result on ADE20K is reported in the paper, why?


**Limitations:**

Please refer to Questions

---

> ### Author Rebuttal · Authors · 2023-08-09
>
> Dear Reviewer W68W,
>
> Thank you for your feedbacks. We will address your concerns below.
>
> **Q1**: Some details are unclear, line 103 says 'the un-masked patches are input into each bottom-up column'. While line 134 and Fig.2 say 'masked image patches are fed into the bottom-up columns and reconstruct unseen patches through top-down column'. Which one is the correct way?
>
> **A1**: Sorry, Line 134 is an obvious mistake that we ignored during proofreading. The correct one should be 'As shown in Figure 2, in MIM pre-training, \textbf{un-masked} image patches are fed into the bottom-up columns and reconstruct unseen patches through top-down columns.' For a typical MIM task, the model receives unmasked patches in the encoder and reconstructs the masked patches (unseen patches) in the decoder. In our pre-training, this scheme remains the same. Thanks for pointing out this mistake and we will correct it in the next version.
>
> **Q2**: Compared with vanilla ViT, the proposed model is quite complex, so I'm wondering about the speed of both the pretrain and inference.
>
> **A2**:  We give a detailed benchmark and analysis of the speed of RevColV2 on the global rebuttal, please refer to it.
>
> **Q3**: Fig.3 compare the MAE decoder with RevColv2 decoder and claims that the proposed decoder can disentangle the low-level and high-level feature better. But for the linear accuracy difference between Level1 and Level4, the difference of MAE is 14.6, while the proposed method is 18.7, the difference seems not significant.
>
> **A3**: Thanks for pointing out this. The difference between the top level and bottom level of the last column is only one aspect of the disentangle learning visualization. As we can see in Figure 3 in paper, more than the difference in the linear probing result of these two levels, the whole distribution of all levels is changed (performance in the left setting first increase then decrease from left to right columns, and the performance in the right setting is gradually increased). We think the whole distribution change is also an important aspect of the representation visualization of RevColV2 models.
>
> **Q4**: PeCo is a strong ImageNet-1k classification baseline while only segmentation result on ADE20K is reported in the paper, why?
>
> **A4**: Yes, PeCo makes really a great job, especially in ImageNet-1k classification. We will append this work to Table 2 in the next version. Specifically, PeCo and RevColV2 achieve comparable results on ImageNet1K dataset only (84.7 vs. 84.5 for base size model and 86.3 vs. 86.5 for large size model).

---

> > ### Comment · Reviewer_W68W · 2023-08-18
> >
> > Thanks for the authors' reply, most of my concerns are resolved, and I keep my score as weak accept

---

### Official Review · Reviewer_yT3P · 2023-07-07

**Soundness:** 3 good
**Presentation:** 3 good
**Contribution:** 3 good
**Rating:** 5
**Confidence:** 5

**Summary:**

This paper proposed novel architecture to explore disentangled representations with masked image modeling. Different from previous MAE-like methods, this paper design a unified network and do not drop the decoder in downstream task. This paper showed that the disentangled representation is learned in different network levels. Experiments are done on ImageNet, MS-COCO and ADE20k for base and large model sizes.

**Strengths:**

1. The idea of combining MIM with disentangled representation learning is novel.
2. The idea of keeping the entire autoencoder tackles the problem of inconsistent representation between pre-training and fine-tuning.
3. The performance in downstream vision tasks is competitive.

**Weaknesses:**

1. Figure 2 is misleading, the pre-training target on ImageNet-1k is single MIM or combined MIM with image labels?
2. The network parameters in Table 1 and Table 2 is not consistent. E.g. for base-size, 101M in Table 1 and 88M in Table 2.
3. As described in line 204, the initialized weight for semantic segmentation is ImageNet-1k classification fine-tuned model and not MIM pre-trained model. It is not fair to compare it with a bunch of MIM pre-trained models.
4. The DropPath op seems to be conflict to the idea of disentangle representation in each level.
5. In the supplementary materials, the linear probing experiments is based on the bottom-up columns which is conflict to the idea of keeping the entire autoencoder.
6. This paper had several typos and grammar mistakes. E.g.
line 1: per-training -> pre-training
line 13: performances -> performance
line 15: intermediately -> intermediate
line 66: per-trained -> pre-trained
line 211: fine-turning -> fine-tuning
7. There are some factual errors in this paper. E.g. in Table 2, the target of ConvNext-B is label not pixel and the target of BEiT-L is DALL-E not pixel.

**Questions:**

Question:
1. Speed. It seems like the training and inference speed of revcol architecture is slower than deeper plain ViT. What is the training and inference speed of RevColV2 in pre-training and fine-tuning?
2. What is the performance of dropping the decoder and only using the encoder for downstream tasks?
3. How to show the scalability of RevColV2?
4. Is there more evidence to show the representation difference of RevColV2 and RevCol? E.g. attention distance, KL divergence between different attention heads, etc.
5. What is the performance of not using DropPath?
6. Why using bottom-up columns representation for linear probing?


**Limitations:**

Yes.

---

> ### Author Rebuttal · Authors · 2023-08-09
>
> Let us answer your questions point by point.
>
> **Q1**: Figure 2 is misleading, the pre-training target on ImageNet-1k is single MIM or combined MIM with image labels?
>
> **A1**: Thanks for pointing out this issue. We do not depict the training task as a single task. Figure 2 illustrates the training target for both pre-training and fine-tuning. We use raw image pixels as targets in self-supervised pre-training, without any labels. All labels are included only in downstream tasks such as image classification. We will add more captions in Figure 2 in the next version.
>
> **Q2**: The network parameters in Table 1 and Table 2 is not consistent. E.g. for base-size, 101M in Table 1 and 88M in Table 2.
>
> **A2**: Tables 1 and 2 illustrate the parameters on different tasks. In Table 1, we show the network parameter on pre-training, which include parameters of both encoder and decoder. In Table 2, we show the parameter of image classification (fine-tuning and inference). As the classification head is applied on top of the last top-down column, only about half of the decoder participates in the calculation (upper triangular of the decoder). And that's why the parameter in Table 2 is lower than in Table 1. We will modify the paper in the revision to make it clear.
>
> **Q3**: The initialized weight for semantic segmentation is ImageNet-1k classification fine-tuned model and not MIM pre-trained model. It is not fair to compare it with a bunch of MIM pre-trained models.
>
> **A3**: Before conducting our experiments, we survey several MIM works including (HiViT, BEiT, ConvNeXt V2). All of them are using ImageNet fine-tuned model weights in the segmentation task. So we follow the settings of these previous MIM works. Indeed, we found image labels can help with the ADE20K segmentation tasks, but for detection, it could harm the performance.
>
> **Q4**: The DropPath op seems to be conflict to the idea of disentangle representation in each level.
>
> **A4**: To my understanding, the concern is about dropping parts of the input images could lead to information loss in propagation. In the training process (only fine-tuning, we do not add Droppath in pre-training), we add the Droppath operation inside each building block, while the shortcut bypass (Identity mapping) still keeps the input image from being discarded in propagation. During the inference process, Droppath does not work, so there is no conflict to feature disentangle.
>
> **Q5**: The linear probing experiments is based on the bottom-up columns which is conflict to the idea of keeping the entire autoencoder. Why using bottom-up columns representation for linear probing?
>
> **A5**: This is a good point. We use the bottom-up columns representation for linear probing because we found it has higher performance. However, it seems like a conflict with the idea of keeping the entire autoencoder. We made some investigation on these phenomena and found that: The reason why the linear probing result of bottom-up column representation is a little higher than the top-down column representation is the depth of the decoder columns. We use a shallower decoder in the default setting, because of the lightweight decoder can be quickly adapted to downstream fine-tuning. But it harms the performance of the linear probing of the decoder representation.
>
> Although this light-weight setting harms the linear probing results of top-down columns, we are more focused on the downstream fine-tuning results in this paper. So we keep this light-weight (shallower) decoder setting by default. Nevertheless, we will report the linear probing of the top-down column representation in the revision. Specifically, the linear probing results of top-down column representations are 64.3\% for base size model and 77.2\% for large size model. These results are still significantly higher than other method in Table 5 in the supplementary material.
>
> **Q6**: This paper had several typos and grammar mistakes. There are some factual errors in this paper.
>
> **A6**: Thanks for pointing out the typos and mistakes. We have already rectify them.
>
> **Q7**: What is the training and inference speed of RevColV2 in pre-training and fine-tuning?
>
> **A7**: We give a detailed benchmark and analysis of the speed of RevColV2 on the global rebuttal mainly focusing on the inference, please refer to it. For training, it will draw the same conclusion with inference. But the speed in training is heavily dependent on the computation resources.
>
> **Q8**: What is the performance of dropping the decoder and only using the encoder for downstream tasks?
>
> **A8**: In the original submission, we conducted a key ablation study that only use the encoder architecture in both pre-training and fine-tuning. To my understanding, you are curious about that only using the encoder architecture in fine-tuning stage (the encoder is also pre-training with the entire auto-encoders). We further conduct experiments under this setting with RevColV2-B, and the experimental results are:
> 83.9\% (-0.8\%) top-1 accuracy on ImageNet1K dataset, 50.4 (-0.7) mIoU on ADE20K dataset. These results are consistent with our original submission and verify the effectiveness of the multi-column decoder design.
>
> **Q9**: How to show the scalability of RevColV2?
>
> **A9**: We investigate the data scaling ability of RevColV2 with the additional CLIP teacher and larger dataset. Please refer to the global rebuttal.
>
> **Q10**: Is there more evidence to show the representation difference of RevColV2 and RevCol?
>
> **A10**: We visualize the attention distance of each self-attention block between RevColv1-ViT-B and RevColv2-B. As shown in the submitted material, the attention distance in MIM pre-trained RevColv2 lies in a more diverse manner compared with the supervised trained RevCol. MIM models often have more diverse attention heads which tend to aggregate both local and global pixels. The conclusion is in accord with <https://arxiv.org/abs/2205.13543>, who did similar visualization on ViT.

---

> > ### Comment · Reviewer_yT3P · 2023-08-19
> >
> > Thanks for the your detailed responses! I will keep my initial score.

---

### Official Review · Reviewer_DHnY · 2023-07-17

**Soundness:** 3 good
**Presentation:** 3 good
**Contribution:** 3 good
**Rating:** 5
**Confidence:** 4

**Summary:**

This paper proposes to keep the entire auto-encoder architecture during both pre-training and fine-tuning based on RevCoI. It contains bottom-up columns and top-down columns, and the information is reversibly propagated and gradually disentangled between them. Better results are achieved on ImageNet-1k and downstream tasks.

**Strengths:**

1. Good motivation to keep the same structure for both pretraining and finetuning.
2. Better results are achieved compared with RevCol.

**Weaknesses:**

1. This paper is a little bit hard to follow, and I do not think the figures help a lot for understanding this paper. Maybe better visualizations/figures are needed.
2. Tiny and small size models are also suggested to exploit. Or the authors needs strong arguments why they did not do so.
3. Some papers achieves better results than this method, but they did not present them and compare with them.

**Questions:**

1. What the performances for tiny and small size model?

---

> ### Author Rebuttal · Authors · 2023-08-09
>
> Dear Reviewer DHnY,
>
> Thank you for your feedbacks. We will address your concerns below.
>
> **Q1**: This paper is a little bit hard to follow, and I do not think the figures help a lot for understanding this paper. Maybe better visualizations/figures are needed.
>
> **A1**: Figure 1 shows the motivation of RevColV2, and Figure 2 shows the overall pipeline of RevColV2 in both pre-training and fine-tuning. We are sorry to cause confusing to you about these illustrations. To better understand the motivation and method of RevColV2, we add a more detailed figure in the additional one page rebuttal material and hope it can address your concerns.
>
> Except this figure, we also made some clear statement of the motivation and main method of RevColV2:
> As shown in Figure 1 (a) in the original submission, traditional supervised method relies on mounts of image labels that learning to project a given image to the label space. However, the label space is a highly compact based on the one-hot scalar label. In other words, this label space under the supervised learning schema the information loss during pre-training. This is not consistent with the purpose of representation learning and will cause sub-optimal performance. Meanwhile, as shown in Figure 1 (b) in the original submission, exists masked autoencoder employs an encoder to embed the masked images into semantic features and a decoder to reconstruct unseen patches. Under this pre-training paradigm, features are rich in low-level information in both input and output. The semantic features, which are desired for downstream tasks, are reserved inside the network. A common method to utilize such semantic feature is to manually partition the encoder and decoder based on the amount of semantic information in features and discard the decoder during downstream fine-tuning. Even so, discarding parts of the pre-trained network could incur information loss when transferring to downstream visual tasks. RevColV2 tackles this problem by keeping the entire autoencoder architecture in both pre-training and fine-tuning. The key component of keeping entire network is separating the low-level and semantic information during the image reconstruction process in pre-training. To accomplish this, we re-design the architecture of RevCol. The new architecture contains a bottom-up reversible column encoder and a top-down reversible column decoder. The bottom-up columns and top-down columns are totally symmetric with masked images and encoder embedding as input. During MIM pre-training, the raw image reconstruction loss is connected to the end of the last column in decoder. Hence low-level information primarily sinks to the bottom level and semantic information moves upwards to other stages based on lossless propagation as shown in Figure 1 (d) in the original submission. For other architecture details, please refer to our updated overall view of RevCol in the global rebuttal material.
>
> **Q2**: Tiny and small size models are also suggested to exploit. Or the authors needs strong arguments why they did not do so.
>
> **A2**: Tiny and small size models are not suitable for MIM training scheme. We seldom see any tiny or small size models in other transformer based MIM works, including but not limited to (MAE, BEiT, CAE, EVA, PeCo, SimMIM, etc.). In paper TinyMIM[1], the author pointed out when the model size is small, MIM pre-training can harm the fine-tuning accuracy on ImageNet-1k classification (Table 2 in [1]). Other methods (TinyMIM/EVA-02) which include extra teacher and use feature distillation in pre-training can make up this defect. Consider our work is trained with purely MIM object, we do not include any tiny or small size models.
>
> [1] Ren, Sucheng, Fangyun Wei, Zheng Zhang, and Han Hu. "TinyMIM: An empirical study of distilling MIM pre-trained models." In Proceedings of the IEEE/CVF Conference on Computer Vision and Pattern Recognition, pp. 3687-3697. 2023.
>
> **Q3**: Some papers achieves better results than this method, but they did not present them and compare with them.
>
> **A3**: Yes, some paper adopt mask distill method and use strong teachers/extra data/different training scheme in pre-training. Consider the space limitation, we cannot include all of them into the paper. We list one representation work here:
> EVA-02 (EVA-02: A Visual Representation for Neon Genesis)
> EVA-series adopt mask distillation pre-training scheme with a 1B parameter EVA-CLIP teacher and use a merged dataset consisting of IN-21K, CC12M, CC3M, COCO, ADE20K, Object365 and OpenImages as pre-training dataset. EVA02 also use ImageNet-21K intermediate supervised fine-tuning.
> We conduct similar experiments which also include mask distillation pre-training scheme with 300M parameter OpenCLIP-ViT-L teacher and pre-trained on Laion-400M dataset. We do not apply any intermediate fine-tuning.
>
> We show the experimental results in Table 1 in the additional one page global rebuttal material.
> Our model does not achieve better performance than EVA-02 because of the inconsistent training settings: 1) Pre-train dataset, 2) Strong teacher, 3) Intermediate fine-tuning. But RevColV2 achieves better performance than all other methods. We give details about the method, training, and results on the global rebuttal, please refer to it.
>
> In the original paper, we do not adopt such training scheme for we hope to use simple training scheme(ImageNet-1k MIM pre-train+ supervise fine-tune) to show the effectiveness of our method for research purpose. Strong teachers, additional data and tricks certainly make strong performance. However, in that case we can not determine whether the progress is attribute to our method itself or other tricks.

---

> > ### Comment · Area_Chair_kShE · 2023-08-17
> > **Read the rebuttal and discuss with authors ASAP**
> >
> > Dear reviewer DHnY,
> >
> > Since you are the only who holds the negative attitude, your opinion is super important. Can you help read the rebuttal ASAP and discuss with the authors and the reviewers. Feel free to raise any new concerns if you have.
> >
> > Best
> > Area Chair

---

> > ### Comment · Reviewer_DHnY · 2023-08-18
> >
> > Thank you for your response. My concerns are all addressed, and I intend to change the rate to Borderline accept.

---

### Official Review · Reviewer_xDT5 · 2023-07-26

**Soundness:** 3 good
**Presentation:** 3 good
**Contribution:** 3 good
**Rating:** 7
**Confidence:** 4

**Summary:**

The paper proposes a revised version of RevCol, referred to as RevColV2, which is applicable for MAE training. RevColV2 consists of an encoder-decoder framework. The encoder is the same as RevCol, while the decoder uses reversed column connections. The paper also uses a unified fine-tuning framework utilizing a decoder for downstream tasks. RevColV2 demonstrates impressive performance on diverse vision tasks.

**Strengths:**

- RevColV2 decoder presents an interesting design with plenty of novelty. In particular, reversed column connection between enc-dec is an innovative architectural approach for MAE training.
- RevColV2 achieves meaningful performance improvements on diverse tasks.
- Multi-column architecture is different from the mainstream of transformer architecture, which enhances the novelty of RevColV2

**Weaknesses:**

- Basic component of the architecture is the same as RevCol. RevColV2 is an improved version, not a new architecture. Although it is interesting, the contribution of V2 paper is limited.
- The paper's contribution is similar to ConvNeXt V2, enhancing existing architecture with MAE training and minor architecture revision. I think shedding light on existing architecture can be a contribution. But, similarity with ConvNeXt V2 paper might decrease the impact of this paper.
- There are no reports for the throughput or latency. FLOPs numbers for detection and segmentation are omitted. I think throughput and FLOPs are necessary for architecture research in 2023. I strongly recommend authors to report those numbers.

**Questions:**

- Although the inverse pass is well described in the main section, I don't know where it is used in the experiments. What is the role of the inverse pass in RevCol V2?
- IN22k intermediate fine-tuning is an interesting case. It would be better if RevCol V2 is compared with IN21k training in the following papers.
  - [1] BEIT V2: Masked Image Modeling with Vector-Quantized Visual Tokenizers
  - [2] DeiT III: Revenge of the ViT

**Limitations:**

.

---

> ### Author Rebuttal · Authors · 2023-08-09
>
> Dear Reviewer xDT5,
>
> Thank you for your valuable feedback. We will address the concerns and answer them below.
>
> **Q1**: Although it is interesting, the contribution of V2 paper is limited.
>
> **A1**: RevColV2 is a new macro design that handles the inconsistent representations between pre-training and fine-tuning by keeping the entire auto-encoders during pre-training and fine-tuning. As you described in the 'Strengths' section, "RevColV2 decoder presents an interesting design with plenty of novelty.", the top-down decoder in RevColV2 is artistically designed to learn disentangled representation in MIM pre-training. We think this new paradigm and its impressive performance are valuable to the research community.
>
> **Q2**: The paper's contribution is similar to ConvNeXt V2, enhancing existing architecture with MAE training and minor architecture revision. I think shedding light on existing architecture can be a contribution. But, similarity with ConvNeXt V2 paper might decrease the impact of this paper.
>
> **A2**: Although both RevColV2 and ConvNeXt V2 are manifested in enhancing existing architecture with MAE training, the motivations behind these are different. ConvNeXt V2 aims to handle the problem of MIM training in modern CNN architecture, while RevColV2 tries to solve the inconsistent representation between pre-training and fine-tuning (it is not mentioned and handled in ConvNeXt V2). Besides, RevColV2 is not only a contributor to the previous version, but also a pioneer to explore disentangled representations in MIM pre-training.
>
> **Q3**: There are no reports for the throughput or latency. FLOPs numbers for detection and segmentation are omitted. I think throughput and FLOPs are necessary for architecture research in 2023. I strongly recommend authors to report those numbers.
>
> **A3**: Thanks for your suggestions. We omitted the FLOPs numbers because we have limited spaces and we found some papers also omitted the FLOPs on downstream (such as HiViT). Nevertheless, we will add this comparison in the further revision.
>
> For example, the FLOPs number on the semantic segmentation task are:
>  | Model    | FLOPs |
>  | ---------- | ----- |
>  | MAE-B      | 2342G |
>  | RevColV2-B | 1008G |
>  | MAE-L      | 4779G |
>  | RevColV2-L | 2990G |
>
> RevColV2 models have lower FLOPs numbers on UperNet semantic segmentation because the number of channels dimension is smaller (the UperHead is more light-weight) and the UperHead contributes a large proportion of FLOPs.
>
> Besides, we give a detailed benchmark of the speed of RevColV2 on the global rebuttal, please refer to that.
>
> **Q4**: Although the inverse pass is well described in the main section, I don't know where it is used in the experiments. What is the role of the inverse pass in RevCol V2?
>
> **A4**: Inverse pass is a basic component of RevColV2 similar to the V1 version. In the pre-training and fine-tuning of RevColV2, we do not save the intermediate feature maps except the last column during the forward pass. When performing backward, we use the inverse pass to recompute the feature maps and gradients of previous columns using the last column features. Thus the memory cost is lower compared with normal training (You can refer to the paper of RevCol v1 for more details). We will make it more clear in the revision.
>
> **Q5**: IN22k intermediate fine-tuning is an interesting case. It would be better if RevCol V2 is compared with IN21k training in the following papers (BEIT V2 and DeiT III).
>
> **A5**: Thank you for your suggestions. BEiT V2 uses vector-quantized visual tokenizers which are trained with an additional CLIP model. We think directly comparing with the CLIP-based method is not fair. For DeiT III which only uses ImageNet dataset without additional knowledge, we will add the comparison to this method. We also show the comparison results here:
>
> ImageNet1K only:
> | Model | Param. | FLOPs | ACC |
> | ----------- | ---- | --- | ---- |
> | DeiT III -B | 87M  | 18G | 83.8 |
> | RevColV2-B  | 88M  | 19G | 84.7 |
> | DeiT III -L | 304M | 62G | 84.9 |
> | RevColV2-L  | 327M | 67M | 86.3 |
>
> ImageNet1K + 22K:
> | Model | Param. | FLOPs | ACC |
> | ----------- | ---- | --- | ---- |
> | DeiT III -B | 87M  | 18G | 85.7 |
> | RevColV2-B  | 88M  | 19G | 86.2 |
> | DeiT III -L | 304M | 62G | 87.0 |
> | RevColV2-L  | 327M | 67M | 87.4 |
>
> We also investigate the data scaling propriety of RevColV2 using an additional CLIP model and compare it with other CLIP-based foundation models. The scaling details and results (including the comparison to BEiT V2 and other counterparts) can be found in the global rebuttal, please refer to it.

---

> > ### Comment · Reviewer_xDT5 · 2023-08-16
> >
> > Thank you for your response.
> > It has addressed some of my concerns.
> >
> > But I still want to know the latency or throughput of RevColV2.
> >
> > Could you provide that information?
> >
> > If the depth of RevColV2 leads to inefficiencies in GPU computation, I would also accept the throughput for a 128-batch on GPU or latency on CPU.
> >
> > I believe throughput/latency comparison is essential for network architecture papers.

---

> > > ### Author Response · Authors · 2023-08-16
> > > **Further response**
> > >
> > > Dear Reviewer xDT5, thanks for your discussion.
> > >
> > > We agree that the analysis of latency and throughput of RevColV2 is essential for network architecture design. In global rebuttal and its additional PDF file, we show the latency of RevColV2 and ViT baseline on a single A100 GPU with batch size 32. The latency of RevColV2 is higher than ViT baseline because of the frequently fragment memory access. Given the same number of FLOPs, the key problem of the low speed is the number of blocks (depth) of RevColV2 models. The original RevCol V1 had realized this phenomenon and we have tried to design a shallower depth network than the V1 version, but still slower than vanilla ViT. Here, we supplement more analysis on the impact of batch size. We show throughput (#image/s) under the different batch size of RevColV2-L and ViT-L on a single A100 GPU.
> > >
> > > ||   bs=16 | bs=32 | bs=64 | bs=128 | bs=256 | bs=512 |
> > > | ---------- | ----- | ----- | ----- | ------ | ------ | ------ |
> > > | RevColV2-L | 432  | 629  | 661  | 697  | 721  | 741  |
> > > | ViT-L   | 730  | 754  | 786  | 811  | 820  | 823  |
> > > | speedup  | 0.591 | 0.834 | 0.841 | 0.859 | 0.879 | 0.900 |
> > >
> > >
> > > The results show that with the increase in batch size, the inference speed gap between RevCol-L and ViT-L is closing because the fragment memory access time can be distributed to each sample. Although the speed of RevColV2 is lower than vanilla ViT, we still think it can be solved by advanced techniques such as kernel fusion and pipeline parallel of multi-column networks. We will add these analysis along with the global rebuttal about speed (throughput and latency) in the next revision. We hope this response can ease your concerns and please let us know if you have any questions.

---

> > > > ### Comment · Reviewer_xDT5 · 2023-08-17
> > > >
> > > > Thank you for your response.
> > > >
> > > > My concerns on throughput and FLOPs have been solved.
> > > >
> > > > I think the GPU memory efficiency during training with the inverse pass (Q4-A4) is a noteworthy advantage of RevCol.
> > > > I recommend that the authors emphasize this in the paper, even though this contribution is the same as RevCol v1.
> > > >
> > > > Thank you for valuable discussion and I will adjust my rating to 'Acceptance'.

---

### Official Review · Reviewer_6koM · 2023-07-26

**Soundness:** 3 good
**Presentation:** 3 good
**Contribution:** 2 fair
**Rating:** 5
**Confidence:** 4

**Summary:**

The paper introduces "RevColv2," an advancement over the RevCol model, enabling compatibility with MIM training. The authors propose the new architecture comprising a bottom-up reversible column encoder and a top-down decoder, facilitating MIM compatibility while preserving disentangled low-level and semantic information throughout the network. The authors conduct experiments on ImageNet, detection, and segmentation tasks and the results show it achieve strong results on ImageNet and ADE20K.

**Strengths:**

•	The idea of RevColv2 to make RevCol compatible with MIM and the design of top-down column decoder is intuitive. It’s nice to see some downstream tasks could benefit from both pre-trained encoder and decoder.

•	The Illustration of key motivation of maintaining disentangled low-level and semantic information is clear and further verified by analysis in Figure 3.


**Weaknesses:**

* The results on ImageNet-1K are strong; however, there is a lack of speed comparison with other methods. It would be valuable to assess the runtime speed for both pre-training and fine-tuning stages, particularly considering the impact of the reversible column network design on speed.
* One benefit for reversible networks is memory-saving (at the cost of some speed). It would be beneficial to discuss whether this holds true for RevColv2. Exploring the trade-off between memory usage and speed for RevColv2 will add valuable insights to the paper.
* In figure 2, the sequence length is different for the same level in the encoder and decoder. It seems unclear about the strategy used to handle this dimension change when connecting the encoder to the decoder.
* For dense prediction tasks, RevColv2 utilizes both encoder and decoder pre-trained weights. To verify the effectiveness of this approach, one missing ablation is to compare to a variant that employs the same encoder and decoder during downstream fine-tuning but only utilizes the encoder's pre-trained weights while initializing the decoder weights randomly.
* While the COCO detection results for the base model are strong, the performance of RevColv2-L appears to lag behind ViTDet-L using Mask R-CNN (54.0 vs. 55.6, citing the results from the ViTDet paper). Additionally, no results for RevColv2-L with Cascade Mask R-CNN are reported. It would be insightful to discussions the scaling results for RevColv2 on detection and provide some intuition on the potential reasons behind the observed performance differences.


**Questions:**

Overall, the authors propose RevColV2 to make RevCol network compatible with MIM training and it achieves consistent better results than RevCol. My main concerns are about some missing analysis/discussions for some ablations and results as listed in the weakness.

**Limitations:**

No limitations have been discussed. Suggestions on potential limitation that worth to discuss could be about the speed issue and further scaling the model.

---

> ### Author Rebuttal · Authors · 2023-08-09
>
> Dear Reviewer 6koM,
>
> Thank you for your valuable feedback. We will address the concerns and answer them below.
>
> **Q1**: The results on ImageNet-1K are strong; however, there is a lack of speed comparison with other methods. It would be valuable to assess the runtime speed for both pre-training and fine-tuning stages, particularly considering the impact of the reversible column network design on speed.
>
> **A1**: We analyze the speed of RevColV2 and the impact of reversible columns in the global rebuttal which mainly focuses on the model inference. As for pre-training and fine-tuning speed, it will draw the same conclusion with which in the global rebuttal. Using reversible in training is similar to checkpoint, which recalculate intermediate activations during backward propagation. Compared to checkpoint, using our reversible backward saves more GPU memory, so that we can use larger batch size to speed up training. In addition, the running time of training is heavily dependent on the computation resources.
>
> **Q2**: One benefit for reversible networks is memory-saving (at the cost of some speed). It would be beneficial to discuss whether this holds true for RevColv2. Exploring the trade-off between memory usage and speed for RevColv2 will add valuable insights to the paper.
>
> **A2**: Yes, this still holds true for RevColV2. Reversible column is the basic component for RevColV2. In the forward pass, we do not need to save the intermediate features. In the inverse pass, we can recompute the feature maps according to the last column outputs. This means the memory cost is significantly lower. In our practice, it only uses about a quarter of the memory cost of the non-reversible counterpart on RevColV2-L. However, due to the feature re-computation and our vanilla implementation, the overall speed that using reversible forward-backward pass and non-reversible forward-backward pass has about the same time consumption on a fixed number of samples during pre-training. But the reversible network support to running on limited computing resources, such as RTX 2080Ti with 11G memory.
>
>
> **Q3**: In figure 2, the sequence length is different for the same level in the encoder and decoder. It seems unclear about the strategy used to handle this dimension change when connecting the encoder to the decoder.
>
> **A3**: For downstream tasks, the sequence lengths between encoder and decoder are the same. For pre-training in Figure 2, these dimensions are different because of the masking strategy. We use the same technique way as MAE to align the sequence length. In the encoder, only visible unmasked patches are used as input. In the decoder the input is the full set of tokens consisting of both encoded visible patches and mask tokens. Line 112 of the original submission had described this practice and we will make it more clear.
>
> **Q4**: To verify the effectiveness of this approach, one missing ablation is to compare to a variant that employs the same encoder and decoder during downstream fine-tuning but only utilizes the encoder's pre-trained weights while initializing the decoder weights randomly.
>
> **A4**: This is a good point. We made experiments that only utilized the encoder's pre-trained weights while initializing the decoder weights randomly with RevColV2-B. This variant achieves 84.4\% (-0.3\%) Top-1 accuracy on ImageNet-1K dataset and 50.7 (-0.6) mIoU on ADE20K dataset, with only ImageNet-1K MIM pre-trained encoder weights. These experimental results draw the same conclusion with the paper that the pre-trained decoder is necessary for RevColV2. We will add this ablation experiment to the paper.
>
> **Q5**: While the COCO detection results for the base model are strong, the performance of RevColv2-L appears to lag behind ViTDet-L using Mask R-CNN (54.0 vs. 55.6, citing the results from the ViTDet paper). Additionally, no results for RevColv2-L with Cascade Mask R-CNN are reported. It would be insightful to discussions the scaling results for RevColv2 on detection and provide some intuition on the potential reasons behind the observed performance differences.
>
> **A5**: In the original submission, we reproduce the results of the ViT backbone on Mask R-CNN detection framework using the different hyper-parameters compared to ViTDet-L due to limited computing resources. Thus the resulting RevColv2-L with Mask R-CNN shows sub-optimal performance.
>
> We investigate the data scaling ability of RevColV2. Specifically, we propose a new learning paradigm for RevColV2 that jointly models the masked image patches (on the top level of the last decoder column) and CLIP features (on the bottom level of the last decoder column) during pre-training. The resulting model shows very impressive results on downstream tasks. More details and results are available in the global rebuttal, please refer to it.

---

> > ### Comment · Reviewer_6koM · 2023-08-15
> > **Response to authors' rebuttal**
> >
> > Thanks to the authors for the rebuttal with additional explanations and experiments. The rebuttal addressed some concerns, but I still have questions regarding Q1/Q2 and Q5.
> >
> > Q1/Q2: The authors provided some inference speed comparison. However, but I remain curious about the unavailability of training speed comparisons. Given that RevColv2 apparently employs the same batch size as MAE (4096), I believe it would be feasible to measure the training speed and memory usage under the same settings. I  think that this comparison would contribute to understand the trade-offs in RevColv2 when comparing with other methods.
> >
> > Q5: It’s nice to see some data scaling results of RevColv2 in Table 1 (rebuttal file). However, Table 1 is more like a system-level comparison as different methods are using different pre-training dataset or teacher model. Conversely, I think a fairer evaluation would be a comparison of models on the same setting, e.g., in Table 4 (the original paper).
> > .

---

> > > ### Author Response · Authors · 2023-08-16
> > > **Further response**
> > >
> > > Dear Reviewer 6koM, thanks for your discussion. We make some further responses to your concerns.
> > >
> > > **Q1/Q2**: The authors provided some inference speed comparisons. However, but I remain curious about the unavailability of training speed comparisons. Given that RevColv2 apparently employs the same batch size as MAE (4096), I believe it would be feasible to measure the training speed and memory usage under the same settings. I think that this comparison would contribute to understand the trade-offs in RevColv2 when comparing with other methods.
> > >
> > > **A1/2**: Thank you for your suggestion. We made some further analysis on the per-training speed and memory cost, and compare it with the popular used ViT-MAE baseline. We take RevColV2-B and ViT-B for comparisons. We test training speed and memory cost on a single A100 (80GB) x 8 machine, with the same dataloader (implemented for our cluster). We use our own codebase for RevColV2 and the official codebase for MAE. The below table shows the real training cost with batch size 4096 for one epoch. To speed up training and save memory, we equip RevColV2 with Flash Attention. We only use data parallel in this testing.
> > >
> > >
> > > | | Time Cost | Memory (each GPU)|
> > > | ----------------------------------- | ---------- | ------ |
> > > | ViT-B | 220s/epoch | 43G |
> > > | RevColV2-B | 249s/epoch | 49G |
> > > | RevColV2-B + FlashAttn | 211s/epoch | 42G |
> > > | RevColV2-B + FlashAttn + Reversible | 240s/epoch | 18G |
> > >
> > > The above table shows that the vanilla implementation of RevColV2 pre-training has a little slower (249s vs. 220s) than ViT. Equipped with FlashAttn, RevColV2 achieves comparable pre-training cost (211s vs. 220s and 42G vs. 43G). We understand that ViT could also be equipped FlashAttn to speed up pre-training. So, we further analyze the impact of reversible propagation. We test the pre-training cost of the Reversible version of RevColV2 (re-compute the intermediate features during backward according to the last column outputs, rather than the vanilla autograd function in PyTorch). Results in the above table show that RevColV2-B can use extremely few GPU memory (only 18G) during pre-training with a total batch size 4096. This allows RevColV2 can be pre-trained with limited resources, such as RTX3090 GPU.
> > >
> > > We will add this analysis to the next reversion and we hope this response can ease your concerns. Please let us know if you have any questions.
> > >
> > > **Q5**: Table 1 is more like a system-level comparison as different methods are using different pre-training dataset or teacher model. Conversely, I think a fairer evaluation would be a comparison of models on the same setting, e.g., in Table 4 (the original paper)
> > >
> > > **A5**: In our investigation of data scaling on RevColV2, we hope to explore the ability upper bound of RevColV2 models. So, we mainly focus on a larger dataset and a strong teacher (this motivation is similar to EVA series). Table 1 is indeed a system-level comparison and shows the ability of RevColV2 models. Behind these experiments, we had made some initial basic experiments in the beginning to verify this new scaling training schema: (1) the same dataset (ImageNet1K), teacher (CLIP-B), and settings (300 epochs) with MaskDistill [1] for RevColV2-B. We validated these schedules on ImageNet1K fine-tuning (2) the pre-training models on Table 1 in global rebuttal, but the exactly the same settings with EVA-02 / ViTDet on COCO detection (1024 x 1024 image size with LSJ argumentations, we use 1536 image size in Table 1 same as EVA-02). We show these results below.
> > >
> > > | Model | pre-training | teachaer | ImageNet1K ft |
> > > | ------------- | ----------------------- | -------- | ------------- |
> > > | MaskDistill-B | ImageNet1K - 300 epochs | CLIP-B | 85.0 |
> > > | RevColV2-B | ImageNet1K - 300 epochs | CLIP-B | 85.5 |
> > >
> > > | Model | image-size | AP |
> > > | ---------- | ---------- | ---- |
> > > | VIT-L | 1024x1024 | 57.6 |
> > > | EVA-02 | 1024x1024 | 59.2 |
> > > | RevColV2-L | 1024x1024 | 59.5 |
> > >
> > >
> > > According to these results, RevColV2-B with the same dataset, teacher, and training settings achieve better performance on ImageNet1K fine-tuning compared with MaskDistill-B. We think this head-to-head comparison verifies the effectiveness of the data scaling with CLIP teacher schema for RevColV2. For results on COCO detection, we think it verifies the ability of data scaling pre-training.  We hope this response can ease your concerns and please let us know if you have any questions.
> > >
> > > [1] Peng, Zhiliang, et al. "A unified view of masked image modeling." arXiv preprint arXiv:2210.10615 (2022).

---

> > > > ### Comment · Reviewer_6koM · 2023-08-19
> > > >
> > > > Thanks for providing the additional results! The results do resolve most of my concerns, although I still feel Table 4 is the simplest setting for ablation or comparison as it does not require the additional teacher for distillation.

---

> > > > > ### Author Response · Authors · 2023-08-19
> > > > > **Feedback to Reviewer 6koM**
> > > > >
> > > > > Dear Reviewer 6koM,
> > > > >
> > > > > We understand that ablations under the exactly same setting can further verify our data scaling method. In the original submission, we use the same settings to verify our method on both Tables 2, 3 and 4.
> > > > > In the investigation of scaling, we show the upper bound (system-level comparison) in the global rebuttal. And we show the head-to-head comparison on ImageNet1K fine-tuning:
> > > > >
> > > > > | Model | pre-training | teacher | ImageNet1K ft acc|
> > > > > | ------------- | ----------------------- | -------- | ------------- |
> > > > > | MaskDistill-B | ImageNet1K - 300 epochs | CLIP-B | 85.0 |
> > > > > | RevColV2-B | ImageNet1K - 300 epochs | CLIP-B | 85.5 |
> > > > >
> > > > > In MaskDistill paper, the authors report the only downstream results on ADE20K with UperNet framework. We use the same setting and run experiments on semantic segmentation. The performance comparisons on ADE20K fine-tuning are shown below.
> > > > >
> > > > > |    | pre-training   | teacher | mIoU | |
> > > > > |---------------|----------------------|---------|------|---|
> > > > > | MaskDistill-B | ImageNet1K-300epochs | CLIP-B | 53.7 | |
> > > > > | RevColV2-B | ImageNet1K-300epochs | CLIP-B | 54.3 | |
> > > > >
> > > > > RevColV2 model with the same pre-training and downstream settings achieves better performance that MaskDistill on both ImageNet-1K and ADE20K. These head-to-head comparisons also verify the effectiveness of our proposed method.
> > > > >
> > > > > We hope our response can ease your concerns about fair comparisons. If you have any questions, please feel free to let us know.

---

### Author Rebuttal · Authors · 2023-08-09

Deal all,

We thank all reviewers' efforts in the comments of our submission. The original review comments recognised our novelty (xDT5, yT3P, W68W) and motivation behind RevColV2 (6koM, DHnY), and acknowledged the performance of RevColV2 (xDT5, yT3P, W68W). While the main concerns of reviewers are focused on the speed (6koM, xDT5, yT3P, W68W), the scaling property of RevColV2 (6koM, yT3P), the additional ablations (6koM, yT3P), the performances (6koM, DHnY) and other detail issues.

In the rebuttal period, we conducted additional experiments and further analyzed the computation cost of RevColV2. We added more comparisons to modern architectures and methods (DeiT III, PeCo, BEiTV2, MaskDistill, EVA02) and additional ablations (encoder only for downstream tasks). We give an analysis of the computation cost for RevColV2. We further investigate the data scaling property of RevColV2 with the help of an additional CLIP model and a larger dataset. For all concerns and questions, we made detailed explanations point-by-point.

We hope our responses can clarify the issues of the reviewers. If there are any other questions, please feel free to ask.

**Global responses:**

**1. Scaling**

We further investigate the scaling propriety of RevColV2 with the help of an additional teacher. The main idea of RevColV2 is learning the disentangled representation during pre-training to keep the entire autoencoders in fine-tuning. This is accomplished by reconstructing the masked image patches on the bottom level of the top-down column decoder. The semantic features are accordingly disentangled to the top levels. We make a further step: explicitly joint-learn masked semantic features on the top-level of the top-down column decoder. Specifically, we use OpenCLIP-L as the teacher to represent the semantic features similar to MaskDistill and EVA.

Except for the additional teacher, we use a larger dataset Laion400M, which contains about 400M unlabeled images in pre-training. Note that we do not use datasets such as COCO, ADE20K, Object365, etc. in pre-training to avoid artificial fitting to specific distribution (this is different from EVA-02 which uses a merged dataset that has overlapped data in the downstream task).

We use 800 ImageNet-1k epochs on Laion400M dataset and then 300 epochs on ImageNet-1k dataset during pre-training. Then we evaluate our model on downstream tasks such as ImageNet1K classification, COCO detection with cascade Mask-RCNN, and ADE20K semantic segmentation with Mask2Former. The newly trained RevColv2-L achieves 87.7\% Top-1 accuracy in ImageNet-1k classification with $224\times224$ input resolution. The larger dataset and the extra teacher lead to better performance compared with purely IN-1k MIM pre-training (86.3\%) and IN-1k MIM + IN-22k intermediate fine-tuning (87.4\%). The performance gain is more prominent on dense prediction tasks. Please see Table 1 of the global rebuttal for more experimental results.

These results verify the data scaling ability of RevColV2, and we hope the RevColV2-L with mask distillation can become a new foundation model in the vision community.

**2. Speed**

We are aware of the current model variants of RevColV2 introduce more latency compared with other works of the similar number of parameters and FLOPs, such as ViT.
We test the inference latency of variant models in Table 2 in the additional page. As described in Speed of [RevColV1](https://openreview.net/forum?id=Oc2vlWU0jFY&noteId=eots0qdyEv)
, fragmented memory access takes a large part of latency. In RevColV2, we made some improvements: 1) remove the up-sample and down-sample operation in RevColV1; 2) reduce the number of total blocks; 3) hard-ware friendly architecture without hierarchy. As shown in Table 2 in the rebuttal PDF file, RevColV2 has lower latency than the V1 version during inference, but is still 1.21x higher than ViT. This is because of the large number of building blocks in RevColV2-L (about twice of ViT-L). Though we reduce the total number of blocks, the multi-column RevColv2 still requires at least 12 blocks in each column in the encoder. Shallower column leads to coarse representation which could harm the performance. On the other hand, if we make the ViT model deeper and maintain the same FLOPs, ViT-L-deeper (48 blocks) and RevColV2-L (48 blocks) have similar latency.

In addition to the above comparison, the fragmented access of memory can be optimized by some techniques which can be further investigated in further work. Here, we give two ways that may be further studied:
- Kernel fusion. This can reduce the frequent access of the memory caused by a large number of blocks.
- Model parallel. Before the calculation of previous columns is finished, parts of the current column can be calculated in parallel. This is the nature of the multi-column network and can be further studied to speed up the inference and training.

---

### Decision · Program_Chairs · 2023-09-21

**Decision:**

Accept (poster)

**Comment:**

This paper investigates the fusion of the RevCol network and MIM pretraining, with a distinctive feature being the retention of the complete auto-encoders during both the pre-training and fine-tuning phases. Following the rebuttal process, it garnered unanimous positive feedback from all five reviewers, with the key novelty being widely acknowledged and appreciated. The authors are encouraged to consolidate all the new results and commitments shown in the rebuttal into the final version.